# Teleportation, Simulation, or Human Video? Data Utilization Law for Robot Manipulation

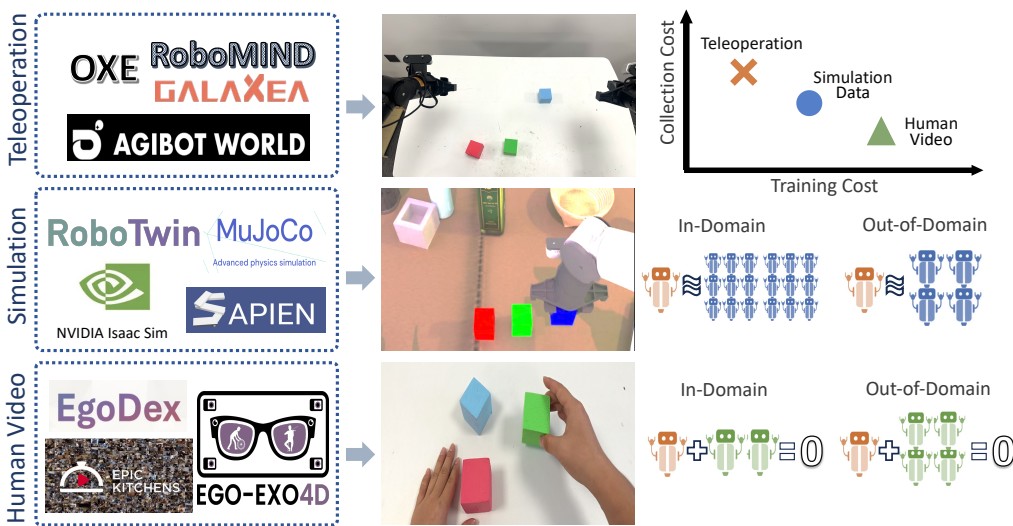

Figure 1: An overview of robotic data sources and their quantified Utilization. The figure illustrates three data types—teleoperation, simulation, and human video—along with their conceptual trade-off between collection cost and training cost.

*"Our biggest cost is not power, or servers, or people. It's lack of utilization. It dominates all other costs."*

— Jeff Bezos

## Abstract

Teleoperation, simulation, and human video represent the three primary data sources for robotic manipulation. Teleoperation data offers high quality at a high collection cost, whereas simulation and human video data are cheaper to acquire but introduce significant embodiment gaps. This trade-off has sparked a debate in the robotics community about the most effective data types for training robot policies. To address this, we introduce a data utilization law for robotic manipulation, drawing an analogy from economics to establish a formal "exchange rate" across data types. We quantify data utilization by using a single real-world teleoperated trajectory as a base unit and then measuring the volume of other data (i.e., simulation or human video) required to achieve equivalent performance. Through a comprehensive investigation across three manipulation tasks—training Diffusion Policy and $\pi_0$ on over 8000 trajectories—we systematically analyze the interplay between real, simulated, and human video data. Our analysis reveals several key findings: 1) Simulation data generally improves model generalization, with an approximate exchange rate of 8 simulation samples providing the equivalent benefit of 1 teleoperated sample. 2) Human video tends to degrade in-domain model performance, where adding approximately 10 human video samples can negate the benefit of a single teleoperated data point. 3) Whether human video helps generalization or simulation aids in-domain performance varies significantly across tasks.

We believe that our work provides crucial insights into balancing the costs of data collection with the computational demands of model training.

# 1 INTRODUCTION

The robotics community has embraced data scaling as a path toward generalizable and robust robot policies, drawing parallels to the recent wave of success in large language models. This pursuit has primarily followed three directions: collecting targeted in-the-lab datasets via teleoperation, creating simulated environments for large-scale data generation, or leveraging unstructured in-the-wild human videos. While the community widely accepts the "more-is-better" paradigm, this heterogeneity of sources means we lack a principled understanding of the relative value of each data type. As illustrated by the data pyramid in Figure 1, roboticists must navigate a trade-off between data fidelity and scale. Teleoperation data, collected directly on a physical robot, offers the highest quality and is most aligned with the test distribution in terms of environment and embodiment. However, this data is expensive and labor-intensive to collect. Conversely, data from simulations and web-scale human videos are far easier to obtain but suffer from significant visual and embodiment gaps. This raises a crucial, yet underexplored, question regarding the true economic efficiency of each data source.

Typically, the cost of data is viewed solely through the lens of collection: Cost total = Cost of data collection. This perspective suggests that cheaper data is always better because it scales more easily. However, this view overlooks a critical factor: data utilization. Some data types are inherently less effective than others. For instance, if a task can be solved by a model trained on 500 teleoperated trajectories, achieving the same performance might require 5,000 human video trajectories. In this scenario, the tenfold increase in training cost could easily outweigh the savings in data collection. This necessitates a more comprehensive cost function: Cost total=Cost data collection+Cost model training. This revised model brings the concept of data utilization to the forefront. In economics, utilization measures how fully a resource is used. Applying this here, our core idea is to establish a quantitative "exchange rate" between different data types, thereby creating a formal method to measure their relative utility and true cost-effectiveness.

To this end, we introduce the data utilization law for robotic manipulation. We operate within the imitation learning paradigm—the dominant approach for real-world robot skill acquisition. We investigate the trade-offs between two critical data pairs: simulation versus teleoperation data and human video versus teleoperation data. To ground our study, we curate a large-scale dataset for a suite of representative manipulation tasks—Pick Dual Bottles, Hand Over Block, and Rank RGB Blocks. This data set comprises real-world teleoperated data collected with an Agilex-2.0 system, RoboTwin 2.0 simulation data Chen et al. (2025), and human video demonstrations processed via MediaPipe Lugaresi et al. (2019). Leveraging state-of-the-art policy models like Diffusion Policy Chi et al. (2023) and pre-trained vision-language-action model $\pi_0$ Black et al. (2024), we meticulously evaluate how varying the composition of training data impacts policy generalization across two key axes: visual background and object position. By analyzing the data utilization ratio, we derive empirical data utilization laws. Our extensive investigation, backed by over 10,000 demonstrations and more than 600 real-world rollouts reveals that:

- We find that simulation data consistently improves a model's ability to generalize to new scenarios. While it is less potent than real-world data, its utility is quantifiable. Our results establish a concrete "exchange rate," revealing that, on average, 8 simulation trajectories provide a generalization benefit equivalent to 1 teleoperated data. This demonstrates that large volumes of cheap simulation data can be a highly effective strategy for enhancing model generalization.

- Conversely, naively adding human video data can be actively detrimental to in-domain task performance. The significant embodiment gap often introduces conflicting signals that confuse the policy. We measured a "negative exchange rate," where adding approximately 10 human video trajectories can completely negate the learning gains from 1 high-quality teleoperated trajectory. This highlights the hidden costs of using easily accessible but poorly aligned data sources for specialized tasks.

- Finally, we find that while human video generally hurts in-domain performance, its effect on generalization can vary from helpful to harmful. Similarly, the benefit of simulation data for in-

domain tasks is inconsistent. This crucial nuance shows that the optimal data-mixing strategy must be tailored to the specific goals and context of the robotic task at hand.

Our work offers a novel framework and a new vocabulary for the systematic analysis of robot data, providing a principled guide for mixing different data types in real-world applications.

## 2 RELATED WORK

**Data scaling in robotic manipulation.** In computer vision and natural language processing (NLP), scaling laws have established that model performance improves predictably with increasing data, model size, and compute (Kaplan et al., 2020; Hoffmann et al., 2022; Devlin et al., 2019; Brown et al., 2020; Deng et al., 2009; Schuhmann et al., 2021; Radford et al., 2021). Robotic manipulation faces unique scaling challenges: collecting real-world interaction data is expensive, time-consuming, and often task-specific. While recent advances leverage simulated interactions (Andrychowicz et al., 2020; Chen et al., 2025; Liu et al., 2023), large-scale teleoperation (O'Neill et al., 2024), or human videos (Hoque et al., 2025; Grauman et al., 2022), existing datasets remain orders of magnitude smaller than those in vision and language. This motivates a shift from merely scaling data volume to systematically understanding data quality and composition.

**Imitation learning in robot learning.** Imitation learning (IL) provides a data-efficient paradigm for training robotic agents by leveraging expert demonstrations rather than trial-and-error exploration. Early IL methods, such as behavioral cloning (BC) (Pomerleau, 1989) and inverse reinforcement learning (IRL) (Ng & Russell, 2000) enabled policy acquisition from demonstration trajectories, but often suffered from compounding errors and poor generalization. More recent approaches combine IL with large-scale pretraining and multimodal inputs, leading to vision-language-action (VLA) models (Brohan et al., 2022; Reed et al., 2022; Black et al., 2024; Bjorck et al., 2025; Wen et al., 2025a;b). These models align perception, language, and action, allowing robots to execute natural-language-specified tasks and transfer across environments.

## 3 APPROACH

### 3.1 DATA SOURCES

We collect data from three primary sources: teleoperation, simulations, and human videos.

**Task description.** We evaluate performance on three robotic manipulation tasks: *Pick Dual Bottles* (easy), *Hand Over Block* (easy), and *Rank RGB Blocks* (hard).

- **Pick Dual Bottles.** The robot uses its right arm to grasp a Sprite bottle, followed by using its left arm to pick up a Coca-Cola bottle. Subsequently, both arms lift the bottles simultaneously in a coordinated manner.
- **Hand Over Block.** It requires the robot's left arm to grasp a red block, perform a hand-over to the right arm, and place the block into the designated top-right goal location.
- **Rank RGB Blocks.** It assesses multi-step rearrangement skills by sorting three colored blocks (Red, Green, Blue) from random permutations into the canonical R–G–B order.

**Teleoperation.** We collected a dataset of expert demonstrations, $\mathcal{D}_{\text{teleop}}$, using two real-world robotic platforms: the Aloha-Agilex-2.0 and the Dual-ARX-R5. For each of the three tasks, we gathered 200 to 500 trajectories per platform, resulting in a total of approximately 1500 teleoperation demonstrations. Both systems utilize a 14-DoF action space, representing the joint angles of their dual 7-DoF arms and binary gripper states. Further details are in Appendix C.1.

**Simulation.** To investigate the role of large-scale, low-cost data, we generated a simulated dataset, $\mathcal{D}_{\text{sim}}$, using the RoboTwin2.0 environment (Figure 2, middle). We created trajectories for both the Aloha-Agilex-2.0 (1,000 trajectories per task for three tasks) and Dual-ARX-X5 (1,000 trajectories per task for two tasks) models, yielding 5,000 simulated demonstrations in total. A key feature of this dataset is extensive **background randomization**, where we systematically varied background textures. See Appendix C.2 for details.

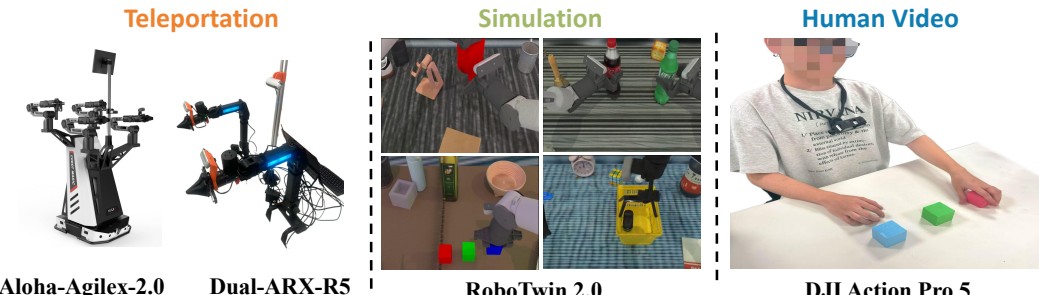

Figure 2: Overview of the three different datasets used in this study: **(Left)** Teleoperation data collected via teleoperation on Aloha-Agilex-2.0 and Dual-ARX-R5; **(Middle)** Simulation data generated in the RoboTwin2.0 environment with background randomization; **(Right)** Human video data captured using a chest-mounted DJI Action Pro 5.

**Human video.** To leverage the natural variability of human motion, we collected a dataset of human demonstrations, denoted as $\mathcal{D}_{\text{human}}$. An operator, wearing a chest-mounted DJI Action Pro 5, performed each task bimanually in the same physical environment as the robot. We recorded 500 trajectories per task, totaling 1,500 demonstrations. We process these videos with MediaPipe (Lugaresi et al., 2019) to extract 3D hand poses. Details are listed in Appendix C.3.

### 3.2 TRAINING STRATEGY

**Policy models and configurations.** Our primary policy network is the Diffusion Policy Chi et al. (2023) equipped with a CNN-based U-Net architecture, where visual features are extracted using a ResNet-50 encoder pretrained on ImageNet. For instruction, we embed textual inputs via Distil-BERT Sanh et al. (2019). We also utilize the state-of-the-art vision-language-action model $\pi_0$. To isolate the effects of the model architecture from the pre-training data, we initialized all models with the PaliGemma pre-trained weights Steiner et al. (2024). This approach allows for a fair assessment of architecture-agnostic generalization. See Appendix D for more details.

**Training setup.** We compare two distinct training protocols. In the **teleoperated-from-scratch** setting, the policy is trained directly and exclusively on the teleoperated demonstration dataset $\mathcal{N}_{\text{tele}}$ for the entire training budget, i.e., $T_{\text{tele}} = T_{\text{total}}$. In the **non-teleoperated pre-train + teleoperated fine-tune** setting, the policy is first trained for $T_{\text{nt}}$ steps on a non-teleoperated dataset $\mathcal{N}_{\text{nt}}$—comprising simulation trajectories $\mathcal{N}_{\text{sim}}$ or human video demonstrations $\mathcal{N}_{\text{human}}$—and then fine-tuned for $T_{\text{tele}}$ steps on $\mathcal{N}_{\text{tele}}$. Since neither simulation-only nor human-video-only training is sufficient to accomplish the tasks, we adopt this pre-train–fine-tune procedure to leverage the complementary strengths of both data sources. To ensure computational fairness, both protocols satisfy $T_{\text{nt}} + T_{\text{tele}} = T_{\text{total}}$, guaranteeing that any performance differences arise solely from the training data composition rather than unequal training effort.

### 3.3 EVALUATION

We conduct all evaluations on a real-world Aloha-Agilex-2.0 platform. To ensure statistical significance, each policy is evaluated for 10 trials per scenario. Performance is measured by the success rate, $\mathcal{S}$, a fine-grained score from 0 to 1 that aggregates the completion of critical sub-steps (e.g., reaching, grasping, placing). Further details on the task setup are provided in Appendix B.2.

We first distinguish between **in-domain** and **out-of-domain** scenarios. In-domain refers to scenes with backgrounds and object positions seen during training, whereas out-of-domain includes changes to either the background or the object positions. As illustrated in Figure 3, we evaluate our policies in three scenarios: **in-domain**, **unseen background**, and **unseen position**. The In Domain scenario replicates the training environment, while Unseen Background and Unseen Position introduce backgrounds and initial object positions not present in the training set, respectively. The teleoperation-only policy is tested in all three scenarios. The simulation-pretrained policy is evaluated in the in-domain and unseen background settings, whereas the policy pretrained with human

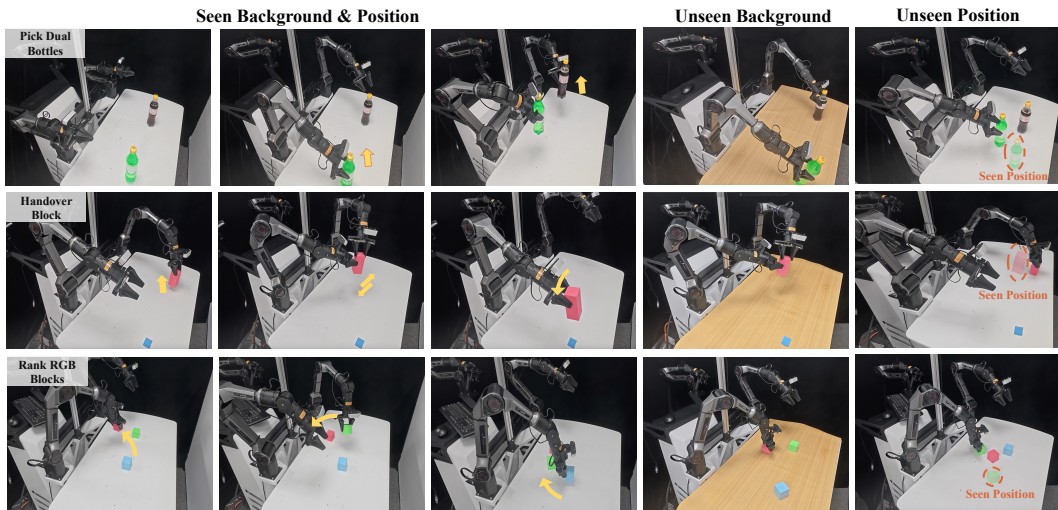

Figure 3: Our three real-world evaluation scenarios: **In-domain (left)**. The training set contains the same background and positions as the evaluation. **Unseen Background (middle)**. The background is new in the test set. **Unseen Position.** The object positions are not in the training set.

| Task | Test Condition | $\mathcal{S}$ (Teleoperation) | $\mathcal{S}$ (+Simulation) | $\eta_{local}$ |
|---|---|---|---|---|
| Pick Dual Bottles | in-domain | 0.70 | 0.55 | −23 |
| | unseen background | 0.20 | 0.30 | 16 |
| Hand Over Block | in-domain | 0.70 | 0.60 | −23 |
| | unseen background | 0.50 | 0.65 | 11 |
| Rank RGB Blocks | in-domain | 0.45 | 0.50 | 18 |
| | unseen background | 0.13 | 0.20 | 4 |

Table 1: Success rate $\mathcal{S}$ and local utilization ratio $\eta_{local}$ when adding simulation data to teleoperation, evaluated under in-domain and unseen-background conditions for three tasks. $\eta_{local}$ measures the relative effectiveness per sample between simulation and teleoperation data.

videos is assessed in the in-domain and Unseen Position settings. Details of the evaluation procedure are provided in Appendix B.3.

## 3.4 DATA UTILIZATION RATIO DEFINITION AND COMPUTATION

We introduce the data utilization ratio, $\eta$, to quantify the value of non-teleoperated data sources ($\mathcal{D}_{sim}$, $\mathcal{D}_{human}$) relative to our primary teleoperated data ($\mathcal{D}_{tele}$). A positive $\eta$ indicates beneficial data, meaning that $\eta$ non-teleoperated samples provide the same performance gain as one teleoperated sample. A negative $\eta$ implies detrimental influence, meaning that $|\eta|$ non-teleoperated samples degrade performance equivalently to one teleoperated sample.

Our first method computes a **local utilization ratio**. We select an operating point $N^*_{tele}$, a quantity of teleoperated data sufficient for the policy to achieve reasonable performance. Near this point, we approximate the task completion metric, $\mathcal{S}$, with a simple additive model:

$$\mathcal{S} = \gamma_{tele}N_{tele} + \gamma_{nt}N_{nt}, \tag{1}$$

where $N_t$ and $N_{nt}$ are the number of teleoperated and non-teleoperated samples, and $\gamma_t$ and $\gamma_{nt}$ are their respective marginal contributions to performance. In practice, we obtain these contributions by running two trainings: one using only $N^*_t$ teleoperated samples, and another using a combination of non-teleoperated pre-training plus teleoperated fine-tuning to the same total sample count. These two measured success rates give us a pair of linear equations that can be solved for $\gamma_t$ and $\gamma_{nt}$. The local utilization ratio is then $\eta_{local} = \gamma_t/\gamma_{nt}$.

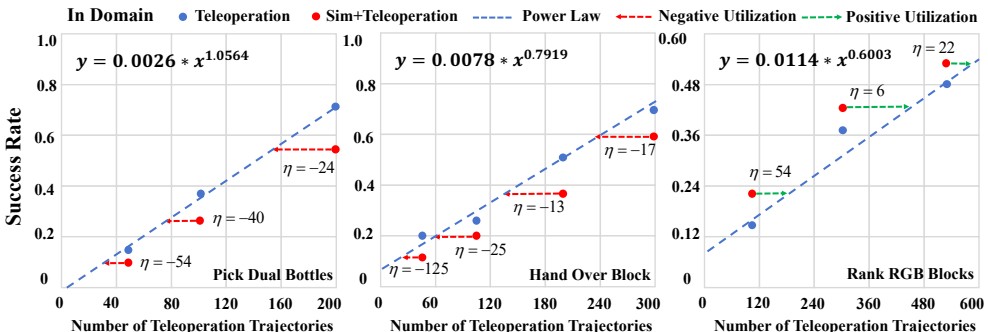

Figure 4: Global utilization ratio of simulation data in in-domain evaluation. All axes are on a logarithmic scale. For simpler tasks (**left and middle**), adding simulation data is detrimental to performance. For a difficult task (**right**), it provides a significant benefit.

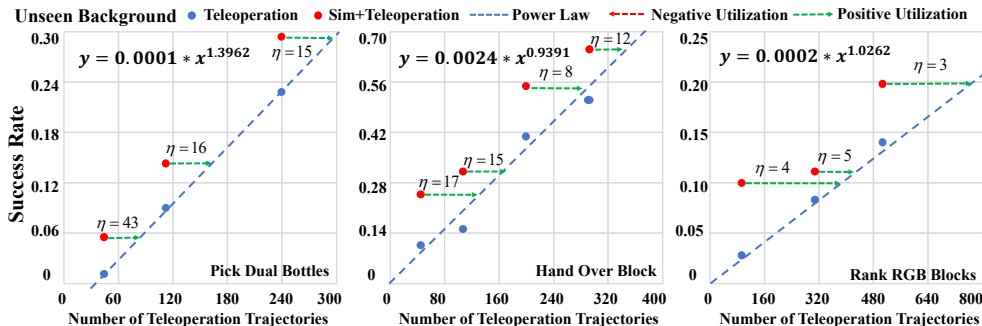

Figure 5: Global utilization ratio of simulation data under out-of-domain background. Simulation data is consistently beneficial across all tasks. Notably, its utility increases with task difficulty.

While this linear approximation is simple and direct, it does not capture the global, sublinear scaling of performance with increasing data, often known as diminishing returns. To obtain a more comprehensive measure that is valid across different data scales, we use a power-law curve to compute the **global utilization ratio**.

First, we fit a power-law function, $\mathcal{S}_{\text{teleop}}(N_t) = \alpha N_t^{\beta}$, to the performance curve using only teleoperated data. Next, we measure the performance $\mathcal{S}_{\text{mix}}$ of a policy trained on a mixed dataset containing $N_t'$ teleoperated and $N_{nt}'$ non-teleoperated samples. We then use the fitted model to calculate the equivalent teleoperation data, $N_{\text{eq}}$, that would be required to achieve this same performance. The global utilization ratio, $\eta_{global}$, is then computed as:

$$N_{\text{eq}} = \left( \frac{\mathcal{S}_{\text{mix}}}{\alpha} \right)^{1/\beta}, \qquad \eta_{global} = \frac{N_{nt}'}{N_{\text{eq}} - N_t'}. \tag{2}$$

$N_{\text{eq}} - N_t'$ represents the contribution of the non-teleoperated data, measured in the currency of equivalent teleoperated samples. However, due to the limited number of points available for fitting the power-law, the resulting fit can be inaccurate. Therefore, we use the global utilization ratio mainly to verify the *sign* (positive or negative utility) of non-teleoperated data across real-robot dataset scales, complementing the more precise local linear estimate near $N_t^*$.

## 4 EXPERIMENTAL ANALYSIS OF DATA UTILIZATION

In this section, we apply the methodology described in Section 3.1 to empirically measure the utilization ratios of simulation and human video data. For each source, we analyze both the local ratio ($\eta_{local}$) to understand its marginal value and the global ratio ($\eta_{global}$) to assess its scalability and effectiveness when integrated at various scales.

| Task | Test Condition | $\mathcal{S}$ (Teleoperation) | $\mathcal{S}$ (+Human Video) | $\eta_{local}$ |
|------|----------------|------------------------------|------------------------------|----------------|
| Pick Dual Bottles | in-domain | 0.70 | 0.45 | $-7$ |
| | unseen position | 0.25 | 0.30 | $+12$ |
| Hand Over Block | in-domain | 0.70 | 0.50 | $-6$ |
| | unseen position | 0.25 | 0.30 | $+8$ |
| Rank RGB Blocks | in-domain | 0.45 | 0.25 | $-2$ |
| | unseen position | 0.17 | 0.13 | $-4$ |

Table 2: Success rate $\mathcal{S}$ and local utilization ratio $\eta_{local}$ when adding human video to teleoperation, evaluated under in-domain and unseen-background conditions for three tasks. $\eta_{local}$ measures the relative effectiveness per sample between simulation and teleoperation data.

## 4.1 EXPERIMENTAL SETUP

We trained baseline policies from scratch on teleoperation data ($\mathcal{D}_{\text{teleop}}$) for each task using different subset sizes: Pick Dual Bottles – 50, 100, 200 demonstrations; Hand Over Block – 50, 100, 200, 300 demonstrations; Rank RGB Blocks – 100, 300, 500 demonstrations. We then evaluated the effect of non-teleoperated data via a pre-train + fine-tune scheme: policies were pre-trained on either 1000 simulation trajectories ($\mathcal{D}_{\text{sim}}$) or 500 human video trajectories ($\mathcal{D}_{\text{human}}$), and subsequently fine-tuned on the same teleop subsets for each task. We evaluate our method under both in-domain and out-of-domain scenarios. In our experiments, the in-domain setting corresponds to scenes with seen backgrounds and positions, whereas the out-of-domain setting corresponds to scenes with unseen backgrounds and positions. Utilization ratios were computed as described in Section 3.4

## 4.2 TELEOPERATION VS. SIMULATION UTILIZATION

We compare simulation and teleoperation data via local and global utilization ratios to evaluate how well simulation can substitute teleoperation in improving task performance in both in-domain and unseen background tasks.

**Local utilization ratio for simulation.** As reported in Table 1, the local utility of simulation data is highly task-dependent. For simpler in-domain tasks (*Pick Dual Bottles*, *Hand Over Block*), simulation exhibits a strong negative utilization ($\eta_{local} = -23$), indicating that the sim-to-real gap introduces detrimental artifacts that harm performance. Conversely, for the complex *Rank RGB Blocks* task, the ratio becomes positive ($\eta_{local} = 18$), suggesting that simulation provides valuable diversity for learning complex coordination. In the unseen background setting, simulation is consistently beneficial, yielding significant positive ratios across all tasks. This confirms that domain randomization in simulation is a highly effective strategy for improving visual robustness.

**Global utilization ratio for simulation.** Our global analysis shows a clear difference in the effect of simulation across tasks. As shown in Figure 4, all axes are shown on a logarithmic scale. The blue dashed line represents a power-law fit to the teleoperation data points. Green and red dashed arrows indicate positive and negative utilization, respectively. The global utilization ratio $\eta_{global}$ is negative for simple in-domain tasks, indicating that adding simulation data reduces performance compared to teleoperation alone. In contrast, for the complex *Rank RGB Blocks* task, $\eta_{global}$ is consistently positive across all data scales, meaning that simulation data improves performance. In the unseen background setting (Figure 5), simulation yields positive utilization for all tasks, suggesting that it is particularly useful for handling background distribution shifts.

## 4.3 TELEOPERATION VS. HUMAN VIDEO UTILIZATION

In this section, we compare human video and teleoperation data via local and global utilization ratios, aiming to assess how well human video can substitute teleoperation in enhancing task performance for both in-domain and unseen position tasks.

**Local utilization ratio for human video.** Human video data shows limited utility and is often detrimental. In-domain, it consistently reduces performance; On simple tasks, seven human videos

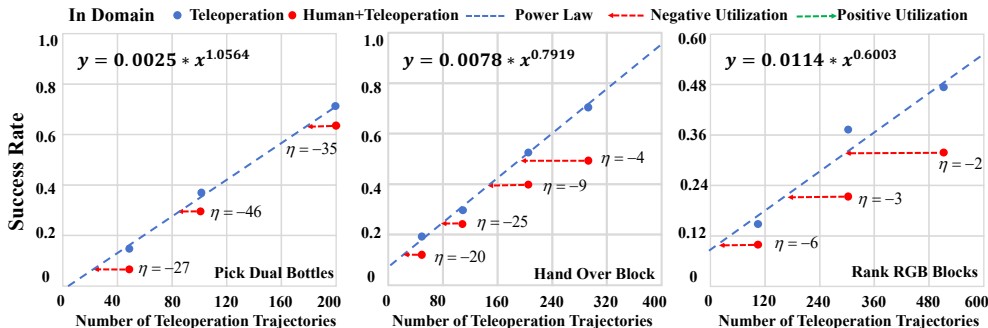

Figure 6: Global utilization ratio of human video data in in-domain evaluation. The addition of human videos results in a negative utilization across all tasks.

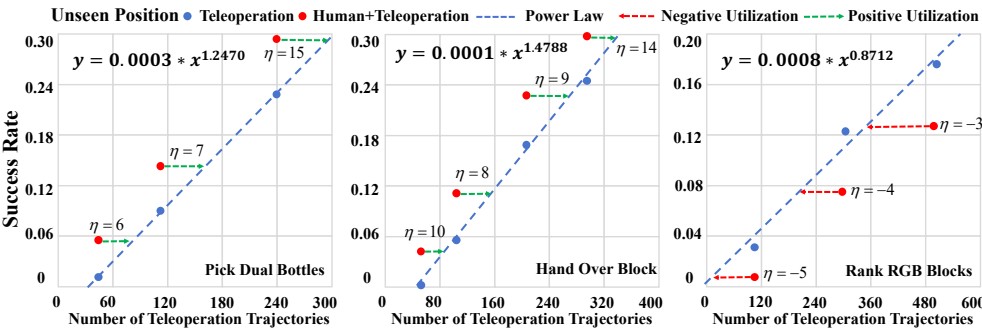

Figure 7: Global utilization ratio of human video data under unseen position: (left, middle)positive for simpler tasks and (right) negative for the difficult task .

cause the same degradation as removing one teleoperated demonstration. Out-of-domain, it benefits simple tasks with a local utilization ratio of 10, but harms the complex task with a ratio of -4.

**Global utilization ratio for human video.** The global analysis reveals that human video data exhibits limited scalability for learning complex robotic skills. For in-domain tasks (Figure 6), the utilization remains negative across all data scales, and the adverse impact increases with task complexity, indicating that the sub-optimal and behaviorally narrow nature of the data can degrade policy performance, particularly in tasks demanding long-horizon reasoning. For simpler unseen position tasks at small scales (Figure 7), a slight positive utilization is observed, but the benefit diminishes as the scale grows. Overall, these results show that human video can provide limited gains in specific cases, yet may introduce biases that reduce generalization in broader settings.

## 5 VERIFICATION OF DATA UTILIZATION LAW

### 5.1 DATA UTILIZATION LAW

To assess the generality of the proposed data utilization law beyond diffusion policy and our primary robot platform, we conducted a controlled evaluation on a distinct policy architecture, $\pi_0$, deployed on the ARX-R5. Two representative tasks were selected: *Hand Over Block* and *Rank RGB Blocks*. The training procedure followed the protocol described in Section 3.2.

As shown in Figure 8, in-domain experiments revealed that adding either simulation or human video data generally decreased performance, with the decline in *Rank RGB Blocks* likely due to $\pi_0$ already achieving high accuracy from teleoperation data alone. In the OOD background setting, simulation data consistently yielded positive utilization, indicating that $\pi_0$ benefits from well-aligned simulated experiences. In contrast, in the OOD position setting, human video data was uniformly detrimen-

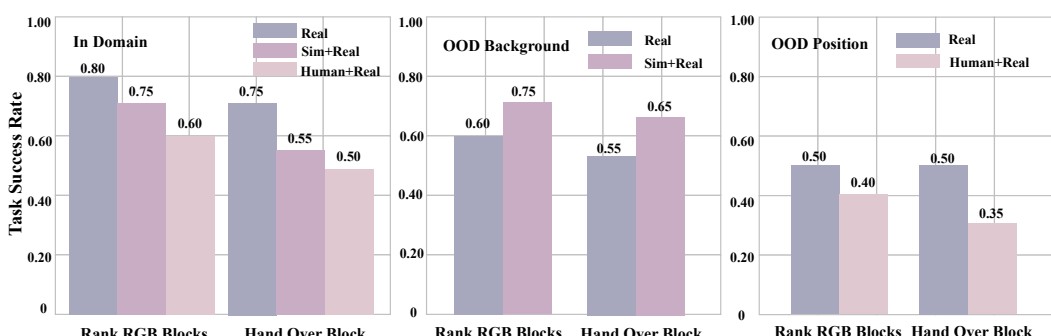

Figure 8: Verification of the Data-Utilization Law on $\pi_0$ and different robot (ARX-R5).

| Strategy | Real Traj. | Sim Traj. | Success Rate |
|---|---|---|---|
| teleoperation-only | 200 | 0 | 0.40 |
| **utilization-guided** | 133 | 2132 | **0.50** |

Table 3: Real-world performance of data collection strategies under a 4-hour time budget. We evaluate all strategies on the *Pick Dual Bottles* task with unseen backgrounds.

tal, suggesting that $\pi_0$ is particularly sensitive to spatial domain mismatches introduced by human demonstrations.

## 5.2 DATA UTILIZATION RATIO FOR BUDGET-AWARE COLLECTION

In the *Pick Dual Bottles* task with an unseen background, we measured that one teleoperation demonstration performs about the same as sixteen simulation demonstrations from the local utilization ratio. Next, we compared data collection speed. In equal collection time, teleoperation can produce only one demonstration, while simulation can produce thirty-two. Since this speed ratio of 32 is greater than the utilization ratio of 16, the simulation is more cost-effective and worth including in a mixed-data strategy.

We conducted a budget-constrained experiment to compare two approaches over a four-hour period (the time required for 200 teleoperation demos). The first approach, a baseline, consisted solely of 200 teleoperated demonstrations. The second, our utilization-guided approach, used a mix of 133 teleoperation and 2,132 simulation demonstrations, calculated using our framework (Appendix B.4). The results are stark (Table 3): our strategy achieved a performance of 0.50, five times higher than the baseline's 0.10. This confirms that a utilization-aware data mixture yields superior real-world performance for the same data collection effort.

## 6 CONCLUSION

We revisited the "more is better" assumption in robot learning by accounting for both collection and training costs and introducing a data utilization law that quantifies exchange rates across data sources. Across representative manipulation tasks, we found that simulation data provides a reliable path to robustness, with roughly 8 simulation trajectories offering the generalization benefit of 1 teleoperated trajectory. In contrast, naively mixing human video can harm in-domain performance, with about 10 human video trajectories offsetting the gain from a single high-quality teleoperated trajectory. These effects are goal-dependent: while simulation generally helps out-of-domain generalization, its in-domain value is inconsistent, and human video's impact on generalization varies. Our framework turns data mixing into an economic design choice, guiding practitioners to prioritize a teleoperation core, augment with scalable simulation, and introduce human video only when alignment mechanisms or broader robustness objectives justify it. We hope this shifts practice from counting samples to valuing them, enabling more cost-effective robot policies at scale.

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

APPENDIX

# A  STATEMENT ON LLM USAGE

In accordance with ICLR's policy on the use of Large Language Models (LLMs) during paper preparation, we declare the extent and nature of LLM involvement in this work. We employed an LLM (*ChatGPT, GPT-5 by OpenAI*) solely for *text refinement* purposes, including improving grammar, enhancing wording clarity, and rephrasing sections to meet academic writing standards. No scientific content, experimental results, or novel ideas were generated by the LLM; all technical contributions—including model design, experiments, and analysis—were conceived, implemented, and verified by the authors.

We have reviewed all text produced with LLM assistance for accuracy and originality, and we take full responsibility for the content of this submission.

# B  EXPERIMENTAL ENVIRONMENT AND IMPLEMENTATION DETAILS

## B.1  HARDWARE SETUP

Our experiments were conducted on the following hardware and software platforms.

- **Computational Platform:** All models were trained and evaluated on a server equipped with $8\times$ NVIDIA H20 GPUs
- **Teleoperation Platforms:** We utilized two distinct bimanual robot setups:
    - **Aloha-Agilex-2.0:** This platform is equipped with three ORBBEC DaBai cameras, providing top-down, left-wrist, and right-wrist views.
    - **Dual-ARX-R5:** This setup uses an Intel RealSense D457 camera for the top-down view and two Intel RealSense D435 cameras for the wrist-mounted views.
- **Human Video Capture:** Human demonstration videos were recorded using a DJI Action Pro 5 camera at a resolution of $1080\times960$ pixels and a frame rate of 30 Hz.

## B.2  DETAILED MANIPULATION TASK DEFINITIONS

We define three bimanual manipulation tasks, each with specific initial conditions, goal states, and success criteria.

- **Pick Dual Bottles:** The task requires the robot to pick up two bottles (e.g., a Coca-Cola and a Sprite bottle, each with specific dimensions and weight) from randomized initial positions within designated zones. The goal is to place them into their respective target regions. The task is scored cumulatively:
    - 0.5 points for successfully grasping the Sprite bottle with the right arm.
    - An additional 0.5 points for grasping the Coca-Cola bottle with the left arm.
    - A full score of 1.0 is awarded only when both bottles are lifted simultaneously to a target height.
- **Hand Over Block:** The agent must first use its left arm to pick up a red block from a specified area (0.5 points). It must then perform a dexterous hand-over, passing the block to the right arm. A full score of 1.0 is awarded upon the right arm successfully placing the block into a designated blue goal region in the top-right corner of the workspace.
- **Rank RGB Blocks:** The workspace contains three blocks of distinct colors: Red (R), Green (G), and Blue (B). Their initial arrangement is randomized among three permutations: (G, R, B), (G, B, R), and (R, B, G). The goal is to use both arms to rearrange the blocks into a specific target sequence (R, G, B) within a designated area. Success is measured by the number of correctly placed blocks, with 0.33 points awarded for each, up to a maximum of 1.0 point.

### B.3 DETAILED EVALUATION SCENARIO CONFIGURATIONS

To rigorously assess the generalization capabilities of our model, we evaluate it across three distinct scenarios, as illustrated in Figure 3.

- **In-Domain (ID):** This scenario replicates the training conditions. The environment features a plain white tabletop background, is free of clutter or distractor objects, and all task-relevant items are initialized within their predefined zones.
- **Out-of-Distribution, Background (OOD-Background):** To test visual robustness, this scenario introduces novel, unseen background textures on the tabletop. These textures are sourced from a diverse library and were not part of the training data.
- **Out-of-Distribution, Position (OOD-Position):** This scenario tests generalization to novel spatial and logical configurations. For all tasks, the initial positions of objects are systematically shifted beyond the boundaries of the in-domain starting zones. For the *Rank RGB Blocks* task, we additionally introduce initial block permutations (e.g., B-R-G, B-G-R) that were never seen during training.

### B.4 OPTIMAL ALLOCATION UNDER TIME BUDGET

We computed the optimal split between real and simulated demonstrations by comparing two factors: (1) the average time required to collect each type of data, and (2) the relative usefulness of each type, measured by the local utilization ratio $\eta$.

First, we measured the collection speed: one real (teleoperated) trajectory took $C_{\text{real}}$ hours to collect, and one simulated trajectory took $C_{\text{sim}}$ hours. The local utilization ratio $\eta$ describes how many simulation trajectories are equivalent to one real trajectory in terms of task success impact.

Given a total time budget $B$, we selected the data mix that maximizes expected real-world performance: - If simulation is both fast to collect and high in relative utility (i.e., speed advantage outweighs lower utility), more simulation data is included. - Otherwise, we rely primarily on real trajectories.

## C DATASET DETAILS

This section provides a comprehensive overview of the data collection, processing, and statistics for the three data sources used in our work: Teleoperation demonstrations, simulated data, and human videos.

### C.1 TELEOPERATION DATA COLLECTION

- **Teleoperation Process:** Expert demonstrations were collected via teleoperation. For the Aloha-Agilex-2.0, we used a symmetric leader-follower setup controlled via the `cobotmagic` interface. For the Dual-ARX_x5, an expert operated the arms using a Virtual Reality (VR) interface. Data from both platforms was recorded at 30 Hz to ensure high-fidelity motion capture.
- **Data Format:** Each timestamped data point comprises three synchronized RGB image streams (640×480 resolution) from the top, left-wrist, and right-wrist cameras, paired with the corresponding 14-dimensional robot action vector in joint space.

A summary of the collected Teleoperation demonstration data is provided in Table 4.

### C.2 SIMULATION DATA GENERATION

- **Simulation Environment:** We generated a large-scale synthetic dataset using our *RoboTwin 2.0* simulation environment, which is built on the [Specify Engine, e.g., MuJoCo or Isaac Gym] physics engine. Physics parameters, such as friction coefficients and gravitational constants, were carefully calibrated to closely mirror real-world dynamics.

Table 4: Statistics of the collected Teleoperation demonstration dataset.

| Platform | Task | # Trajectories | Avg. Length (Steps) | Total Steps |
|---|---|---|---|---|
| Aloha-Agilex-2.0 | Pick Dual Bottles | 200 | 450 | 45,000 |
| | Hand Over Block | 300 | 300 | 30,000 |
| | Rank RGB Blocks | 500 | 600 | 60,000 |
| Dual-ARX_x5 | Hand Over Block | 200 | 350 | 42,000 |
| | Rank RGB Blocks | 300 | 650 | 78,000 |

- **Domain Randomization:** To foster the learning of robust representations, we applied extensive domain randomization across several axes:
  - **Background Textures:** Tabletop and background wall textures were randomly sampled from a large library, including procedurally generated patterns and images from public datasets [Specify source, e.g., Poly Haven].
  - **Lighting Conditions:** The position, color, and intensity of multiple light sources in the scene were randomized for each episode.
  - **Distractor Objects:** To simulate a cluttered environment, we randomly placed a variety of distractor objects (with randomized shapes, sizes, and colors) on the tabletop, avoiding the primary task interaction areas.

The statistics for the generated simulation data are summarized in Table 5.

Table 5: Statistics of the generated simulation dataset.

| Task | # Trajectories | Avg. Length (Steps) | Total Steps |
|---|---|---|---|
| Pick Dual Bottles | 10,00 | 450 | 4,500,00 |
| Hand Over Block | 10,00 | 300 | 3,000,00 |
| Rank RGB Blocks | 10,00 | 600 | 6,000,00 |

## C.3 HUMAN VIDEO DATA COLLECTION AND PROCESSING

- **Collection Protocol:** Human demonstrators performed the manipulation tasks on a white tabletop that matched the robot's workspace. A camera was chest-mounted to capture a first-person-like viewpoint. Demonstrators were instructed to perform actions clearly and to minimize hand-object occlusions to ensure high-quality data.
- **MediaPipe Processing Pipeline:** We processed the raw video footage using the MediaPipe framework (Lugaresi et al., 2019) to extract bimanual hand poses. The pipeline consists of the following steps:
  1. For each video frame, we apply MediaPipe's hand tracking solution to detect and extract the 3D coordinates of 21 keypoints for each hand.
  2. The raw 3D keypoint data for both hands is then normalized [Describe normalization, e.g., relative to the wrist's position and palm size] and concatenated.
  3. This process yields a final 48-dimensional action vector representing the full bimanual pose for each frame, which serves as the action signal for our policy.
- **Data Visualization:** Figure **??** illustrates our data processing pipeline, showing an original video frame alongside the corresponding 3D hand pose skeleton extracted by MediaPipe.

## D POLICY TRAINING

### D.1 MODEL ARCHITECTURES

**Base Backbones.** We employ two distinct policy backbones to examine architecture-agnostic generalization. Our primary backbone is the Diffusion Policy Chi et al. (2023), implemented using a

CNN-based U-Net architecture. Visual observations are processed via a ResNet-50 encoder pre-trained on ImageNet, producing spatial feature maps that are projected into the latent space of the U-Net. Language-conditioned inputs are embedded using DistilBERT Sanh et al. (2019), whose outputs are concatenated with visual latents prior to the denoising stage.

The secondary backbone, denoted as $\pi_0$, is initialized with PaliGemma pre-trained weights Steiner et al. (2024). Unlike Diffusion Policy's explicit multimodal fusion layers, $\pi_0$ adopts a transformer-based joint vision-language encoder, enabling native handling of mixed-modality inputs through shared cross-attention mechanisms.

**Human-Video Pretraining.** For cross-domain transfer, both models are initially trained on a large-scale human demonstration dataset (*human video*), where action trajectories are represented in a 48-dimensional continuous space. In Diffusion Policy, the pretrained weights derived from human-video are selectively transferred to the Teleoperation fine-tuning stage to address the dimensional mismatch between the 48-D pretraining action space and the 14-D robot control interface. Specifically, layers that depend directly on the input action dimensionality—such as the `combine` layers and the final noise-prediction head—are *excluded* from weight loading and instead randomly initialized. This selective loading strategy allows the pretrained visual and language representations to be retained, while ensuring that action-dependent parameters are correctly adapted to the robot's lower-dimensional control space.

In contrast, $\pi_0$ handles dimensional compatibility more natively. We set the *maximum action dimension* to 48 during initialization, such that inputs from both human-video (48-D) and robot control (e.g., 14-D) domains are automatically padded to the unified length of 48 before entering the model. This configuration avoids architectural modifications and enables direct usage of pretrained weights across domains.

## D.2 TRAINING HYPERPARAMETERS

Table 6: Training Hyperparameters for Diffusion Policy and $\pi_0$.

| Hyperparameter | Diffusion Policy | $\pi_0$ |
|---|---|---|
| Optimizer | AdamW | AdamW |
| Learning Rate | $1 \times 10^{-4}$ | $2 \times 10^{-5}$ |
| Weight Decay | 0.01 | 0.01 |
| Batch Size | 768 | 64 |
| Total Training Steps ($T_{\text{total}}$) | 13000 | 40000 |
| Pre-training Steps ($T_{\text{pre}}$) | 10,000 | 30,000 |
| Fine-tuning Steps ($T_{\text{ft}}$) | 3000 | 10,000 |

