# OpenReview forum: "Teleportation, Simulation, or Human Video? Data Utilization Law for Robot Manipulation"
_ICLR.cc/2026/Conference — ICLR 2026 Conference Withdrawn Submission_

### Official Review · Reviewer_WJbm · 2025-10-31

**Soundness:** 2
**Presentation:** 2
**Contribution:** 3
**Rating:** 4
**Confidence:** 3

**Summary:**

The paper studies how three common data sources for robot manipulation, teleoperation on real robots, simulation rollouts, and in-the-wild human videos, translate into actual policy performance when training imitation-learning models. It proposes a data utilization law or an empirical exchange-rate framework that measures how many samples of one source are needed to match the benefit of one teleoperated trajectory. Using Diffusion Policy and $\pi_0$ across several real-world manipulation tasks and generalization settings, the authors find that simulation is consistently helpful but weaker per sample, while naively adding human video can hurt in-domain performance, though its effect on out-of-domain generalization varies by task.

**Strengths:**

- Performing data curation is important for the robotic imitation learning problem. The paper proposed an empirical data utilization law that turns a messy multi-source training question into measuring quantitative exchange rates.

- The paper used the proposed exchange rate to design empirical studies across different data modalities, tasks, and policy backbones, which provided practical findings potentially guide the data collection procedure.

**Weaknesses:**

- The data utilization law is calibrated on specific architectures, datasets, and training recipes (e.g., diffusion or flow-based policies). Importantly, the exchange rates may shift under different visual representations or learning algorithms.

- The exchange rates offer limited practical guidance on how to extrapolate data collection with different data scales, tasks, or domain shifts.

- The motivations and formulations of local and global utilization ratios, as well as their correspondence to experiment results, are not explained clearly. See the question part for details.

- Since the paper focuses on empirical studies, including preliminary open-source code and checkpoints would strengthen the conclusions.

**Questions:**

- Fig 1: The rightmost subfigures in the second and third (in-domain vs out-of-domain) are confusing? Can you explain what they mean?

- Sec 3.4 line 256 and line 267: there are confounding factors in both the dataset and training procedure: (1) the optimality of the data in simulation and human videos, (2) the fine-tuning dataset and performance gain from this teleoperated fine-tuning.

-  Sec 3.4: What is the intuition for the local and global utilization ratios? What is the difference between them? Specifically, why do we choose the linear function and power law function?

- Table 1 and Table 2: It is not clear how to get the local utilization ratios from Eq 1 using numbers in the tables.

- Fig 4, Fig 5, Fig 6, and Fig 7: These figures are confusing. From the figures, both x and y axes are on a linear scale (instead of a logarithmic scale) but the power law curve is on logarithmic scale. So the power law curve should go through zero. Can you explain how the numbers correspond to the quantities in Eq. 2?

- Sec 4.2: Why does the in-domain simulation data decrease the performance while the out-of-domain simulation data increase the performance? If I understand correctly, the simulated task only randomizes the background of the scenes. In this case, it looks like the background is irrelevant information and the underlying transition function is the same.

- Any discussion on the practical guidance for data collection from the data utilization law?

---

### Official Review · Reviewer_vrK1 · 2025-11-03

**Soundness:** 2
**Presentation:** 2
**Contribution:** 2
**Rating:** 2
**Confidence:** 3

**Summary:**

The paper studies the relative utility of different data sources (teleoperated data, simulation, human video) for robotic manipulation.

**Strengths:**

The paper studies an important question that would be of interest to the community.

**Weaknesses:**

- It would be helpful to provide the motivation for selecting the methodology used in the study (selection of tasks, data quantities, etc.). It is a bit hard to understand why these settings were chosen and what makes them well suited for studying this phenomenon.
- It is not clear how robust the findings of the study are. The number of trials is relatively small and many of the experiments use few data samples. It would be good to discuss and analyze this.
- The experiments are performed in fairly limited settings and then used to make very broad claims and "laws". I do not think that the current experiments substantiate the broad claims.
- There are a number of confounding factors that can impact the results. For example, if using video data does not lead to an improvement it is hard to know if that is due to the nature of video data or the specific method used to leverage video data.

**Questions:**

Please see the weaknesses above.

---

### Official Review · Reviewer_PQyE · 2025-11-05

**Soundness:** 2
**Presentation:** 3
**Contribution:** 2
**Rating:** 2
**Confidence:** 4

**Summary:**

This paper investigates how different data sources - real robot teleoperation, simulation, and human video demonstrations - contribute to training effective manipulation policies, and introduces a “data utilization law” to quantify their relative value. The authors define a formal exchange rate: how many samples of one data type equal one real teleoperated sample in terms of performance. Using over 8,000 trajectories and two state-of-the-art policy models across three bimanual manipulation tasks, they find that simulation data generally improves out-of-domain generalization, with roughly 8 simulation trajectories being equivalent to 1 teleoperated trajectory, though it sometimes hurts in-domain performance. In contrast, human video data often harms robot learning, with approximately 10 human video samples negating the benefit of one teleop sample, though rare task-specific cases show small gains for generalization. Overall, the key takeaway is that not all data is equally useful: effective robot scaling requires valuing data, not just counting it - mixing simulation strategically with a teleoperation core, and using human video only under strict alignment, leads to more cost-effective and robust robot policies.

**Strengths:**

1. The paper addresses a critical scaling challenge in robot learning - quantifying the relative value of teleop, sim, and human-video data - with a clear practical impact on data-collection strategies.
2. The paper is well-written, logically organized, and easy to understand, with clear motivation, methodology, and visualizations.
3. Uses state-of-the-art architectures (Diffusion Policy and pi0), increasing the relevance and credibility of the findings.

**Weaknesses:**

1. The paper does not evaluate simulation-pretrained policies on novel object-position shifts nor human-video-pretrained policies on novel backgrounds. This leaves a gap in understanding whether the observed utilization ratios hold under all forms of distribution shift, and whether some data modalities may still benefit complementary generalization dimensions.

2. It is not clarified whether the camera intrinsics and extrinsics between simulation and real-world robotic setups were precisely matched. Without such calibration, the reported sim-to-real performance may partially reflect camera-space discrepancies rather than intrinsic data utility, potentially understating simulation effectiveness.
3. The significant drop in performance when using human-video pretraining may stem from hand–eye coordination and embodiment mismatches between human demonstrations and robot kinematics/observations. The paper does not analyze or control for this gap, which may limit the interpretation of the negative utilization results for human video.
4. The study only considers a pretrain-then-finetune pipeline and does not evaluate co-training (joint training on real + simulated or real + human-video data). Prior work has shown co-training to reduce domain gaps, so the absence of such baselines may understate the potential value of mixed-source data.
5. Recent works have demonstrated performance gains from combining teleoperation and human-video data. The paper does not discuss why those methods succeed where this paper observes negative transfer — e.g., differences in data preprocessing, action-space alignment, observation modalities, or architectural mechanisms. Without this comparison, the conclusions regarding human-video utility may appear overgeneralized.
6. Only 10 trials were done per task, which might lack statistical significance.

Other minor issues include:

1. There is a typo in the title and Figure 2: teleportation → teleoperation
2. Broken figure ref in line 748
3. Line 744: seems like an incomplete todo.

**Questions:**

1. Why did the authors not try testing sim-pretraining on novel object positions, and human-video pretraining on novel backgrounds?
2. For sim-pretrained policies, were the camera extrinsics/intrinsics in sim and real matched?
3. Could the hand-eye coordination gap between real datasets and human videos be a reason for the starkly bad performance of human-video-pretrained policies?
4. Did the authors try co-training policies on the two datasets, real and sim/human-video (similar to [1] and [2])?
5. There are recent papers that have shown that co-training policies with teleop and human-video data does improve performance, e.g. [3]. Can the authors comment on the experimental setup requirements that were potentially met in those papers, which are potentially not met in this paper?
6. How much does the human-video pre-training performance rely on the


References:

[1] What Matters in Learning from Large-Scale Datasets for Robot Manipulation, Saxena et al., 2025

[2] Sim-and-Real Co-Training: A Simple Recipe for Vision-Based Robotic Manipulation, Maddukuri et al., 2025

[3] EgoMimic: Scaling Imitation Learning via Egocentric Video, Kareer et al., 2024

---

### Official Review · Reviewer_hcdA · 2025-11-06

**Soundness:** 2
**Presentation:** 1
**Contribution:** 2
**Rating:** 2
**Confidence:** 4

**Summary:**

This paper investigates the relative utility of different data sources—teleoperated robot demonstrations, simulation-generated data, and human video—for training robot manipulation policies. The authors conduct a study across three distinct robot manipulation tasks, generating a considerable volume of training data for each. Their core contribution is the introduction of a new concept: the Data Exchange Rate, which quantifies the relationship and transfer effectiveness between these different data modalities. The research attempts to derive "Data Utilization Laws" for robot manipulation based on their empirical findings.

**Strengths:**

1. **Focus on Data Utilization**: The study correctly identifies and addresses the crucial importance of data utilization strategies and data source analysis in the field of robot manipulation, which is a vital area of research for scaling up robotic capabilities.

2. **Substantial Data Collection and Novel Concept**: The authors undertook the effort to collect a substantial amount of training data across the three defined manipulation tasks and introduced the "Data Exchange Rate" concept, offering a new quantitative metric to assess data transfer effectiveness.

3. **Comprehensive Modality Scope**: By simultaneously investigating data from teleoperation (robot in-domain), simulation, and human video (out-of-domain), the paper covers the three most prevalent and debated data sources in embodied AI research, providing a broad comparative view of their respective roles.

**Weaknesses:**

1. **Insufficient Related Work Survey**: The related works section is remarkably sparse and lacks depth. It fails to acknowledge or discuss a significant body of contemporary research that has already explored the concept of data scaling laws and data modality transfer in robot manipulation [1][2][3], making it impossible to ascertain the novelty or differences between the proposed work and existing studies.

2. **Obvious and Limited Conclusions**: The derived "new conclusions" often reiterate findings that are already established consensus or widely observed heuristics within the embodied robotics community. The paper does not yield genuinely novel or insightful conclusions that fundamentally advance the understanding of data utility beyond current general knowledge.

3. **Methodology as Experimental Setup**: The so-called "core method" of the paper is essentially a description of simple experimental configurations and conditions. The presentation style strongly resembles a technical report of empirical findings rather than the introduction of a novel framework or technical contribution, which significantly limits its impact as a research paper.

4. **Poor and Hasty Manuscript Quality**: The manuscript exhibits significant signs of being rushed and poorly proofread. There are multiple instances of inconsistent notation (e.g., three different symbol conventions for "demonstrations" across sections 3.1, 3.2, and 3.4), inconsistency between symbols in Formula 1 and the accompanying text, a clearly unrelated prompt-like text snippet in Appendix C.3, and missing/placeholder data visualizations. This lack of professionalism severely degrades the paper's readability and credibility.

5. **Limited Task Scope for Generalization**: The conclusions are based on experiments across only three specific, low-complexity manipulation tasks. This narrow scope is insufficient to generalize the proposed "Data Utilization Laws" to the broader, high-dimensional, and long-horizon challenges typical of advanced robot manipulation.

```
[1] ManiBox: Enhancing Spatial Grasping Generalization via Scalable Simulation Data Generation, arXiv 2024.11
[2] Data Scaling Laws in Imitation Learning for Robotic Manipulation, ICLR 2025
[3] Is Diversity All You Need for Scalable Robotic Manipulation?,  arXiv 2025.07
```

**Questions:**

1. **Data Scaling Law Literature**: Which specific, existing research papers on robot data scaling laws and multi-modality transfer were reviewed, and how does the "Data Exchange Rate" differ fundamentally from their quantitative metrics?

2. **Generalization to Complex Tasks**: How would the proposed "Data Utilization Laws" change or apply when the experiments are extended to long-horizon, multi-step, or bimanual manipulation tasks?

3. **Conclusion Insight**: What is the single, most counter-intuitive or novel finding from this study that definitively goes beyond the existing common knowledge in robot manipulation data?

4. **Notation and Writing**: Will the entire manuscript be thoroughly revised to ensure consistent notation, correct all grammatical and typographical errors, and integrate all missing figures and appendix content?

---

### Note · Authors · 2025-11-13

I have read and agree with the venue's withdrawal policy on behalf of myself and my co-authors.